# Automatic Morphology Control of Liquid Metal using a Combined Electrochemical and Feedback Control Approach

**DOI:** 10.3390/mi10030209

**Published:** 2019-03-26

**Authors:** Ming Li, Hisham Mohamed Cassim Mohamed Anver, Yuxin Zhang, Shi-Yang Tang, Weihua Li

**Affiliations:** 1School of Engineering, Macquarie University, Sydney, NSW 2122, Australia; ming.li@mq.edu.au; 2School of Mechanical, Materials, Mechatronic and Biomedical Engineering, University of Wollongong, Wollongong, NSW 2522, Australia; hmcma667@uowmail.edu.au (H.M.C.M.A.); yz472@uowmail.edu.au (Y.Z.)

**Keywords:** liquid metal, morphology control, electrochemistry, feedback control system

## Abstract

Gallium-based liquid metal alloys have been attracting attention from both industry and academia as soft, deformable, reconfigurable and multifunctional materials in microfluidic, electronic and electromagnetic devices. Although various technologies have been explored to control the morphology of liquid metals, there is still a lack of methods that can achieve precise morphological control over a free-standing liquid metal droplet without the use of mechanical confinement. Electrochemical manipulation can be relatively easy to apply to liquid metals, but there is a need for techniques that can enable automatic and precise control. Here, we investigate the use of an electrochemical technique combined with a feedback control system to automatically and precisely control the morphology of a free-standing liquid metal droplet in a sodium hydroxide solution. We establish a proof-of-concept platform controlled by a microcontroller to demonstrate the reconfiguration of a liquid metal droplet to desired patterns. We expect that this method will be further developed to realize future reconfigurable liquid metal-enabled soft robots.

## 1. Introduction

Gallium-based liquid metal alloys, such as Eutectic Gallium Indium (EGaIn, 75% gallium and 25% indium) [1] and Galinstan (68.5% gallium, 21.5% indium, and 10% tin) [2], have been gaining momentum in recent years as promising soft-matter electronics and multifunctional materials [3,4,5]. This is due to their high electrical and thermal conductivities, high surface tension, extremely low vapour pressure and melting points below room temperature [6,7]. Combining the unique properties of both liquid and metal, gallium-based liquid metals provide unprecedented properties for a wide range of applications, including prosthetics, soft robotics and reconfigurable circuits [3], which cannot be achieved using conventional solid materials. In comparison to mercury (Hg), the most commonly known liquid metal, liquid metal alloys based on gallium have much lower toxicity [8], making them much safer to use for research and commercial purposes. Moreover, unlike mercury, gallium-based liquid metal alloys can form a thin oxide layer on the surface [9], enabling these metals to be patterned or re-configured into useful shapes, such as micro- or nano-sized droplets [10,11,12,13].

The capability to manipulate the morphology of liquid metals containing gallium is useful for various applications [3,14,15], including switches, reconfigurable antennas, metamaterials and plasmonics. Among different methods utilized to change the shape of the liquid metals, e.g., mechanical [16], pneumatical [17], and continuous electrowetting [18,19,20,21] techniques, electrochemistry [22,23,24,25] may be one of the most popular methods due to its ease of operation and manipulation, simplicity of the instrumentation, and low cost. Although the shape of liquid metals can be manipulated by electrochemical deposition or removal of the oxide layer on its surface to decrease or increase the interfacial tension, precise automatic control over the morphology is still challenging. Moreover, previous electrochemical control of liquid metal morphology is always confined within enclosed microchannels. 

In order to overcome the aforementioned challenge, we proposed the use of a feedback control system combined with electrochemical techniques to manipulate the morphology of an EGaIn droplet within sodium hydroxide (NaOH) solution in a simple, fast, automatic and precise manner. We established a feedback control system consisting of a microcontroller, a current sensor and a switching circuit. We also optimized the parameters, such as electrochemical oxidation potential and NaOH concentration, which affect the morphological control behaviours. To the best of our knowledge, it is the first time that electrochemistry together with a feedback control system for the morphologic control over liquid metals has been reported.

## 2. Results and Discussion

When a moderate voltage is applied to the liquid metal droplet, electrochemical oxidation occurs on the surface of the metal, which can significantly decrease the interfacial tension [24]. Despite the oxide normally acting as a physical barrier to flow, the metal can flow in the presence of a base, which competes with the electrochemical deposition by dissolving the oxide [19,24]. Here we studied the control over the morphology of liquid metals using electrochemical techniques together with a feedback control system. The experimental setup is shown in Figure 1a; the control system has three main parts: a microcontroller module (Arduino Mega 2560, Arduino, Mansfield, TX, USA), a current sensing module, and a switching circuit module. The proof-of-concept platform used for controlling the morphology of the EGaIn droplet consisted of two laser-patterned acrylic sheets. The bottom plate has a chamber with a diameter and depth of 50 and 6 mm, respectively. The top plate has twelve small holes (diameter of 1 mm) arranged in a circle for accommodating electrodes. Two plates were assembled together using four M4 screws, and the chamber contains the NaOH solution, an EGaIn droplet and a pair of electrodes (Figure 1a upper-right corner). We used a function generator (33250A, Agilent, Santa Clara, CA, USA) to provide a direct current (DC) voltage. A relatively low DC voltage of 5 V was used to ensure that the EGaIn droplet could flow smoothly from the anode at the centre of the chamber towards the cathodes at the sidewall. If a lower voltage was applied, the EGaIn droplet could not reach the cathodes but flattened down and stopped moving instead. Also, we chose NaOH at a concentration of 0.4 M to allow the EGaIn droplet to move smoothly without significant morphology distortion. A thick oxide layer formed on the surface of the EGaIn droplet during the experiment at a lower NaOH concentration, which prevented the EGaIn droplet from moving towards the cathodes. While at a higher concentration, unnecessarily electrolysis occurred at the electrodes.

Figure 1b illustrates the scheme of the feedback control system. An Arduino Mega 2560 module was chosen as the controller in the system. The controller works by (1) first taking in the current value detected between the EGaIn droplet and one of the cathodes using the current sensing circuitry (feedback signal), (2) comparing it with the current value detected between the EGaIn droplet and the other cathode (reference signal), and (3) determining the pulse width of the pulse width modulation (PWM) control signal via an N-channel power metal oxide semiconductor field effect transistor (MOSFET, STMicroelectronics BUZ71, Geneva, Switzerland). One Ω shunt resistors were connected in series with each electrode to sense the current applied to the EGaIn droplet, and an RC filter circuit was developed to convert the PWM voltage signal obtained across the shunt resistors to analogue signals. We next used a dual operational amplifier chip (Texas Instruments LM358p, Dallas, TX, USA) to amplify the filtered voltage signal, which ensured that the values interpreted by the controller were of high accuracy. Under the combined effect of electrochemistry and feedback control, we aimed to move the EGaIn droplet (volume of 1 mL), initially in the centre of the chamber, towards the two cathodes arranged differently along the chamber sidewalls to form two protrusions (see Figure 1c). Such a feedback control strategy attempts to balance the currents between each of the protrusion–cathode pairs to form symmetric patterns. 

The EGaIn droplet was demonstrated to be re-configured automatically and precisely to patterns with two main protrusions that intersected at five different angles: 180°, 150°, 120°, 90° and 60° (see Appendix A). We could see the formation and elongation of liquid metal protrusions towards the cathode. The elongation of the liquid metal is due to the non-uniform oxidation of the liquid metal at the areas facing towards the cathode, and this oxidation process can locally reduce its interfacial tension [24]. Therefore, this may generate a Marangoni flow travelling towards the area with a higher interfacial tension (i.e., the centre of the chamber), which drags the liquid metal protrusions towards the cathode. Figure 2 represents a series of snapshot images showing the variations in EGaIn droplet morphology within 12 seconds after activating the control system. Before the experiment, the levelling of the platform was carefully adjusted by rotating the M4 screws to remove the gravity-induced morphology change. Also, calibration was performed by comparing current values measured by a multimeter and obtained by the controller. The configurations of the EGaIn droplet are relatively stable and can maintain their morphology as long as the control system is activated. We found that the EGaIn droplet could not form stable protrusions if its volume is less than 300 µL. 

Additionally, the limits of the developed system were explored by placing the cathodes at a smaller angle of 30° (see Appendix A). As shown in Figure 3a, the EGaIn droplet reached the cathodes after activating the system for 3 seconds, a “Y” shaped pattern was observed instead of a “V” shape, and the shape could be maintained. As the angle between the cathodes reduces, the interference between the two cathodes increases, leading to a less stable pattern. In order to investigate the repeatability of the system, three separate experiments were conducted to change the morphology of the EGaIn droplet to a pattern with the angle between the two protrusions of 60°. The photographs showing the morphology of the EGaIn droplet at 8 s after activating the system are represented in Figure 3b. EGaIn droplet morphology was almost the same in the three separate experiments, indicating that the system has a relatively high repeatability and high accuracy. This system is also compatible with more than two electrodes. Figure 3c shows the control of EGaIn droplet morphology using three cathodes, in which a fan-shaped pattern with three protrusions was generated. The scheme of the feedback control system is similar to the case given in Figure 1b, however, in this case the reference signal is the smallest current value detected between the EGaIn droplet and the other two cathodes. It is expected that this system can create more complex shapes using more electrodes. 

Finally, we examined the capability of our technique of the dynamic control over the morphology of the liquid metal droplet. By changing the applied potential to different cathodes, we demonstrated that our technique is able to dynamically transform the patterned protrusions with an intersecting angle of 180° to new patterns with two different intersecting angles: 120° (Figure 4a) and 60° (Figure 4b). We also examined the leaching of the gallium and indium ions within the NaOH solution using inductively coupled plasma mass spectrometry (ICP-MS, Thermo Scientific™ Neptune XT™, Waltham, MA, USA). Two tests were performed before and after conducting the morphological transformation experiments of the liquid metal droplet 100 times. Our results indicate that gallium can be dissolved into the NaOH solution, with the concentration increasing from ~1.1 to 766.4 μmol/L.

## 3. Conclusions

In summary, we have described a method to automatically control the morphology of a droplet of EGaIn liquid metal using electrochemistry and a feedback control system. The ability to tune the interfacial tension of the EGaIn droplet with moderate voltages allows the droplet to be deformed into different shapes. By integrating with a feedback control system, automatic and precise control over the morphology of the EGaIn droplet has been demonstrated. We have achieved the reconfiguration of the EGaIn droplet to patterns with two protrusions that intersect at five different angles, i.e., 60°, 90°, 120°, 150° and 180°. We found that the EGaIn droplet reached the cathodes after activating the control system for 3 seconds, and maintained its morphology as long as the system was activated within 12 seconds. We also showed that our technique is able to achieve dynamic control over the intersecting angle between the two protrusions of the EGaIn droplet. We expect that this method would enable stretchable and reconfigurable components with precise control that can be widely used in Micro-Electro-Mechanical Systems (MEMS) and soft robotics. 

## Figures and Tables

**Figure 1 micromachines-10-00209-f001:**
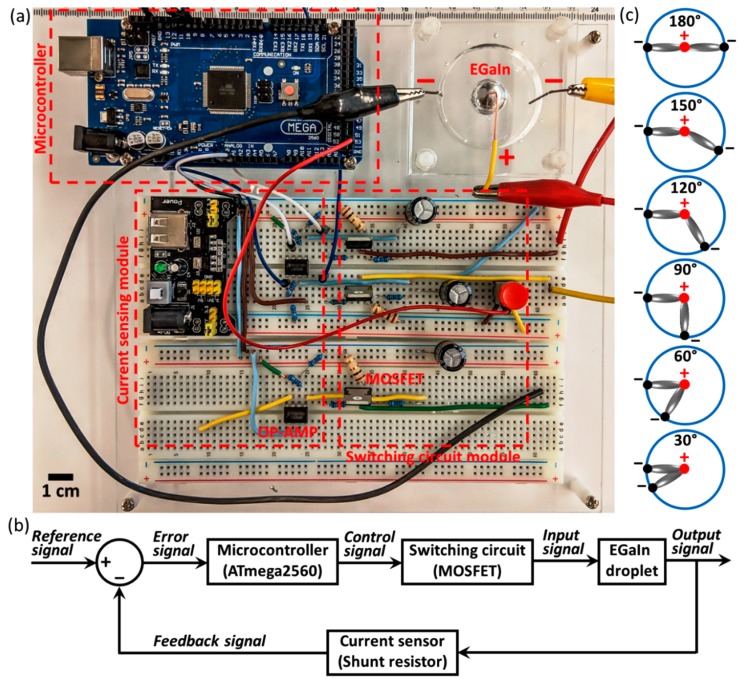
(**a**) A photograph of the experimental setup. (**b**) A scheme of the feedback control system. (**c**) Schematics illustrating the proposed morphological control of the EGaIn droplet using six different configurations of electrodes.

**Figure 2 micromachines-10-00209-f002:**
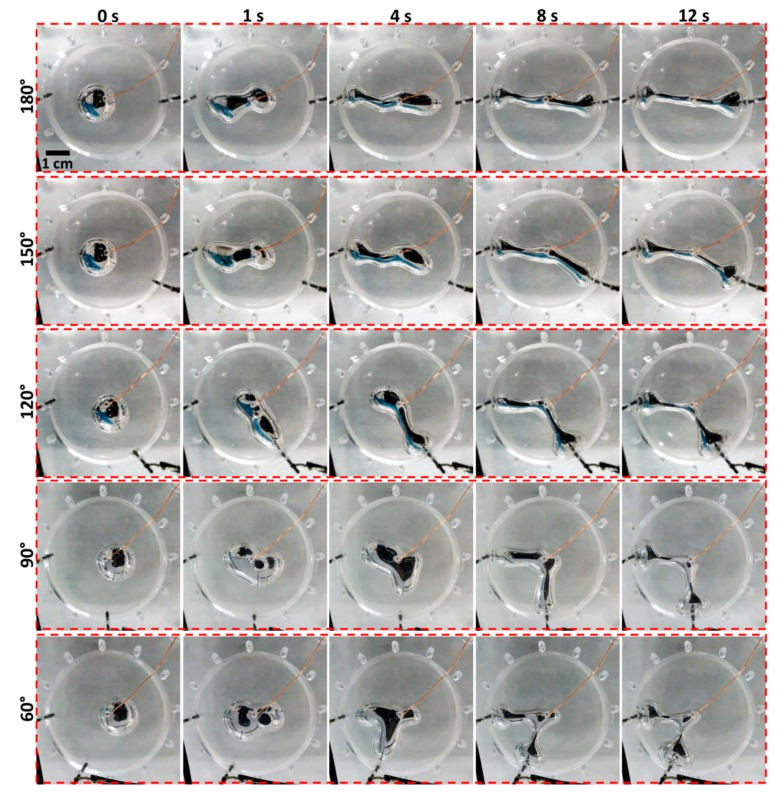
Automatic control of the morphology of the EGaIn droplet to five desired patterns (angles of protrusions are 180°, 150°, 120°, 90° and 60° from top to bottom) within 12 seconds after activating the control system. A series of snapshot images captured at 0, 1, 4, 8 and 12 s are shown for each pattern.

**Figure 3 micromachines-10-00209-f003:**
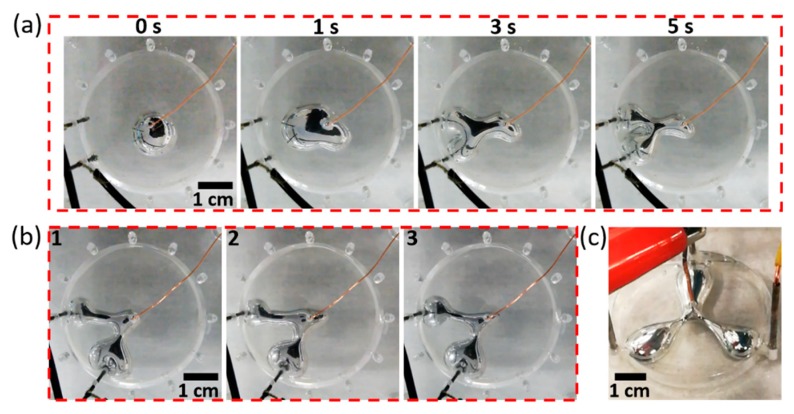
(**a**) Exploring the limit of the system by arranging two cathodes to a configuration of 30°. A series of snapshot images captured at 0, 1, 3 and 5 s after activating the system are shown. (**b**) Exploring the repeatability of the system by recording the morphology variations of the EGaIn droplet at 8 s after activating the system in three separate experiments. (**c**) Morphological control of the EGaIn droplet using three cathodes to yield a fan-shaped pattern with three protrusions.

**Figure 4 micromachines-10-00209-f004:**
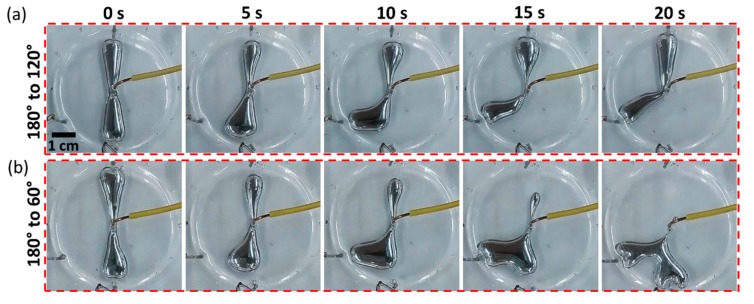
Exploring the dynamic morphological control of the EGaIn droplet by changing the angle between the patterned protrusions from 180° to (**a**) 120°, and (**b**) 60°. A series of snapshot images captured at 0, 5, 10, 15 and 20 s after applying potential to different cathodes are shown.

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
