# Peer review of "Automatic Morphology Control of Liquid Metal using a Combined Electrochemical and Feedback Control Approach"

_micromachines, 2019, doi:10.3390/mi10030209_

Round 1

Reviewer 1 Report

This is a very strong work

The authors explored electrochemistry and used an Arduino controlled feedback unit to change the morphology of liquid metal droplets an ionic solution

Just a few minor comments

There are a few typos in the abstract that should be fixed

“A Gallium-based liquid metal alloys have been” to “Gallium-based liquid metal alloys have been”

“Electrochemical manipulation can be relatively easy to perform,” to “Electrochemical manipulation can be relatively easy to apply on liquid metals,”

“but calls for” to “but there are needs for”

“We expect that this method could be “ to “We expect that this method will be”

Introduction

Add the following references about the applications and properties of liquid metals such as catalytic conversion and surface oxides “Nature Communications 10 (1), 865, 2019 and 8, 14482, 2017 ”

Change “N-Channel power Metal Oxide Semiconductor Field Effect Transistor” to “N-channel power metal oxide semiconductor field effect transistor”

“obtained across the shunt resistor to an analogue signal” to “obtained across the shunt resistors to analogue signals”

I have a equation – considering the shape in Fig 3 c – Can the author change the shape to a constant propeller movement by quickly alternating the current and voltage. The authors do not include this in the paper if this is too difficult. How fast liquid metal can change shape ? This is optional

Conclusion: Include extra parametric outcomes of the paper in the conclusion.

Author Response

Response to Review 1 comments

This is a very strong work.

The authors explored electrochemistry and used an Arduino controlled feedback unit to change the morphology of liquid metal droplets an ionic solution.

Just a few minor comments

Comment 1: There are a few typos in the abstract that should be fixed

“A Gallium-based liquid metal alloys have been” to “Gallium-based liquid metal alloys have been”

“Electrochemical manipulation can be relatively easy to perform,” to “Electrochemical manipulation can be relatively easy to apply on liquid metals,”

“but calls for” to “but there are needs for”

“We expect that this method could be “ to “We expect that this method will be”

Response 1: We thank the reviewer for finding these typos. We have corrected them in the revised manuscript.

Introduction

Comment 2:  Add the following references about the applications and properties of liquid metals such as catalytic conversion and surface oxides “Nature Communications 10 (1), 865, 2019 and 8, 14482, 2017 ”

Response 2: We thank the reviewer for pointing out these two important references about the applications and properties of liquid metals. We have added both references in the revised manuscript.

Comment 3: Change “N-Channel power Metal Oxide Semiconductor Field Effect Transistor” to “N-channel power metal oxide semiconductor field effect transistor”

“obtained across the shunt resistor to an analogue signal” to “obtained across the shunt resistors to analogue signals”

Response 3: We thank the reviewer for the findings. We have corrected them in the revised manuscript.

Comment 4: I have a equation – considering the shape in Fig 3 c – Can the author change the shape to a constant propeller movement by quickly alternating the current and voltage. The authors do not include this in the paper if this is too difficult. How fast liquid metal can change shape ? This is optional

Response 4: We thank the reviewer for raising this insightful question regarding morphology control. Yes, it is possible to change the shape to a constant propeller movement by quickly alternating the applied potential. We have conducted a proof-of-concept experiment to show that our technique is capable for dynamic morphology control. As shown in Figure 4 in revised manuscript, the patterned protrusions with an intersecting angle of 180° can change to new patterns with two different intersecting angles: 120° (Fig. 4a) and 60° (Fig. 4b) by changing the applied potential of different cathodes.

Conclusion

Comment 5: Include extra parametric outcomes of the paper in the conclusion.

Response 5: We thank the reviewer for indicating this missing detail, which can provide the reader with more information about the features of our approach. We have added the following text in the revised manuscript.

Page 67, Line 162-165

We found out that the EGaIn droplet reached the cathodes after activating the control system for 3 seconds, and maintained its morphology as long as the system is activated within 12 seconds. We also showed that our techniques is able to realize the dynamic control over the intersecting angle between the two protrusions of the EGaIn droplet.

Reviewer 2 Report

This manuscript reports on controlling the direction of liquid metal actuation via an electrochemical approach where 2 cathodes are used with a feedback loop. The topic is certainly of interest and to date the direction of liquid metal actuation has been demonstrated in one direction and the ability to control movement in different directions and develop interesting shapes is new using this approach. There are a few points however which should be considered.

Can the system be reversed? Once a particular shape is attained can it quickly be changed back to the starting position?

The size of the cathode will play a significant role – it appears that the tip of a wire is only immersed in the electrolyte, which is much smaller than the pool of liquid metal which is the anode – this will have implications for maintaining the current and also the driving force that can be achieved – was the size of the cathode varied at any stage?

It would be beneficial to check if changing shapes is possible – say for Figure 1 c can the topmost 180 degree shape be transformed into the 60 degree shape by changing the applied potential to different cathodes or would it need to go back to the starting position first?

How much leaching of liquid metal occurs into solution?

Author Response

Response to Review 2 comments

This manuscript reports on controlling the direction of liquid metal actuation via an electrochemical approach where 2 cathodes are used with a feedback loop. The topic is certainly of interest and to date the direction of liquid metal actuation has been demonstrated in one direction and the ability to control movement in different directions and develop interesting shapes is new using this approach. There are a few points however which should be considered.

Comment 1: Can the system be reversed? Once a particular shape is attained can it quickly be changed back to the starting position?

Response 1: As long as the system is deactivated, the sodium hydroxide (NaOH) solution will remove the oxide layer on the surface of the liquid metal and the patterned droplet will change back to its original spherical shape.

Comment 2: The size of the cathode will play a significant role – it appears that the tip of a wire is only immersed in the electrolyte, which is much smaller than the pool of liquid metal which is the anode – this will have implications for maintaining the current and also the driving force that can be achieved – was the size of the cathode varied at any stage?

Response 2: The depth of the NaOH solution was set at ~6 mm and we always fully immersed the tips of the electrodes into the solution, therefore, the area of the cathodes in contact with the NaOH solution remained constant in all experiments.

Comment 3: It would be beneficial to check if changing shapes is possible – say for Figure 1 c can the topmost 180 degree shape be transformed into the 60 degree shape by changing the applied potential to different cathodes or would it need to go back to the starting position first?

Response 3: Per the reviewer’s comments, we conducted additional experiments to show the change of the angle by applying electrical potential to different cathodes. We have added the following text and Figure 4 about dynamic morphological control into the revised manuscript:

Page 5, Line 142

Finally, we examined the capability of our technique for the dynamic control over the morphology of the liquid metal droplet. By changing the applied potential to different cathodes, we demonstrated that our technique is able to dynamically transform the patterned protrusions with an intersecting angle of 180° to new patterns with two different intersecting angles: 120° (Fig. 4a) and 60° (Fig. 4b).

Comment 4: How much leaching of liquid metal occurs into solution?

Response 4: Per the reviewer’s comment, we measured the leaching of liquid metal using inductively coupled plasma mass spectrometry (ICP-MS). We have added the flowing text regarding leading of the liquid metal in the revised manuscript.

Page 5, Line 146

We also examined the leaching of the gallium and indium ions within the NaOH solution using inductively coupled plasma mass spectrometry (ICP-MS). Two tests were performed before and after conducting the morphological transformation experiments of the liquid metal droplet for 100 times, respectively. Our results indicate that gallium can be dissolved into the NaOH solution, with the concentration increased from ~1.1 to 766.4 μmol/L.

Reviewer 3 Report

It is an interesting and well-presented paper, which I suggest to be accepted for publication.

I only found 2 typo items which do not require an additional review process:

- page 2 ,line 5 from bottom: "concentration" instead of "concertation"

- page 5: notation and reference of subfigures in Fig. 3 fluctuates a bit, e.g., reference is made to Fig. 3B in the text, but the figure is introduced as Fig. 3(b). In addition, it took me a moment to find Fig. 3C - perhaps the notation (c) right outside the figure could be added.

Author Response

Response to Review 3 comments

It is an interesting and well-presented paper, which I suggest to be accepted for publication.

I only found 2 typo items which do not require an additional review process:

Comment 1: page 2 ,line 5 from bottom: "concentration" instead of "concertation"

Response 1: We thank the reviewer for finding this typo. We have corrected it in the revised manuscript.

Comment 2: page 5: notation and reference of subfigures in Fig. 3 fluctuates a bit, e.g., reference is made to Fig. 3B in the text, but the figure is introduced as Fig. 3(b). In addition, it took me a moment to find Fig. 3C - perhaps the notation (c) right outside the figure could be added.

Response 2: We thank the reviewer for this comment, which help us to represent data in a clear manner. We have introduced subfigures in Fig. 3 as Fig. 3a, Fig. 3b, and Fig. 3c in the revised manuscript and added the notation (c) right outside the figure as the reviewer suggested.   
